# SceneDecorator:
# Towards Scene-Oriented Story Generation with Scene Planning and Scene Consistency

**Quanjian Song**[1, *†], **Donghao Zhou**[2, †‡], **Jingyu Lin**[1, †‡], **Fei Shen**[3],
**Jiaze Wang**[2], **Xiaowei Hu**[4, §], **Cunjian Chen**[1, §], **Pheng-Ann Heng**[2]

[1]Monash University    [2]The Chinese University of Hong Kong
[3]National University of Singapore    [4]South China University of Technology
Project Page: `https://lulupig12138.github.io/SceneDecorator`

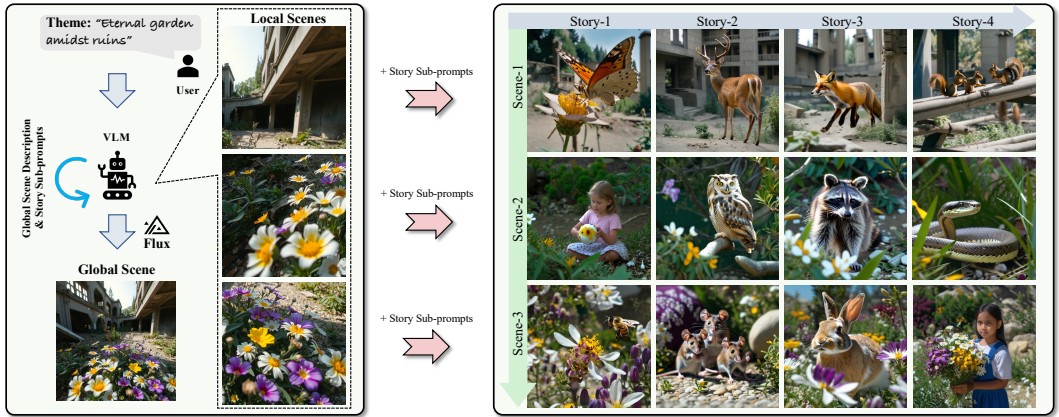

Figure 1: Overview of SceneDecorator. SceneDecorator manages to "decorate" the scenes of story images, ensuring narrative coherence across different scenes (*green arrow*) and scene consistency across different stories (*blue arrow*), all based on a concise user-provided theme.

## Abstract

Recent text-to-image models have revolutionized image generation, but they still struggle with maintaining concept consistency across generated images. While existing works focus on character consistency, they often overlook the crucial role of scenes in storytelling, which restricts their creativity in practice. This paper introduces *scene-oriented story generation*, addressing two key challenges: (i) *scene planning*, where current methods fail to ensure scene-level narrative coherence by relying solely on text descriptions, and (ii) *scene consistency*, which remains largely unexplored in terms of maintaining scene consistency across multiple stories. We propose *SceneDecorator*, a training-free framework that employs *VLM-Guided Scene Planning* to ensure narrative coherence across different scenes in a "global-to-local" manner, and *Long-Term Scene-Sharing Attention* to maintain long-term scene consistency and subject diversity across generated stories. Extensive experiments demonstrate the superior performance of SceneDecorator, highlighting its potential to unleash creativity in the fields of arts, films, and games.

---

*Work done at Monash University.

†Equal contribution.

‡Project leaders.    §Corresponding authors.

39th Conference on Neural Information Processing Systems (NeurIPS 2025).

# 1 Introduction

Text-to-image (T2I) models [1, 2, 3, 4] have demonstrated impressive proficiency in generating high-quality images from text descriptions. However, they struggle to maintain concept consistency across generated images due to their stochastic nature [5]. Such consistency holds significant commercial value and application potential in education [6], art [7, 8], and entertainment [9], underscoring the need for the task of story generation that can create multiple images with consistent concepts [10, 11].

Considering the significance of story generation, numerous prior studies have been devoted to advancing this important task. Early studies such as PorotoSV [12] and FlintstonesSV [13] are typically trained on given datasets with consistent characters. These methods achieve decent performance in specific domains but are inherently limited in generalization. Leveraging the exceptional generation quality of diffusion models, subsequent works [5, 14, 15, 16, 17] have begun exploring open-domain characters. This advancement achieves a compelling balance between realism and aesthetics.

Although existing story generation methods have made significant progress in character consistency, they usually overly focus on preserving characters while neglecting scene depiction, which is equally crucial for conveying the narrative of stories [18]. In light of that, the motivation for this paper arises: *How can we achieve story generation from the perspective of scenes?* In this work, we formulate *scene-oriented story generation*, which presents two primary challenges: (i) *Scene planning*: Existing approaches generate the scenes of story images solely based on text descriptions, leading to a lack of scene-level narrative coherence. This coherence also plays a vital role in enhancing storytelling visual fluency. (ii) *Scene consistency*: In practical scenarios like film storyboarding [19], it is crucial to generate diverse story images with consistent scenes that align with different plots and characters. Maintaining long-term scene consistency across multiple stories remains underexplored.

This paper introduce **SceneDecorator**, a training-free framework for scene-oriented story generation (see Figure 1), aimed at addressing the above challenges. SceneDecorator contains two key techniques:

(i) To tackle *scene planning*, we develop a **VLM-Guided Scene Planning** strategy. It utilize the visual perception of Vision-Language-Model (VLM) to create scenes and story sub-prompts in a "global-to-local" manner. Specifically, this strategy begins with a VLM that interprets the user-provided theme to generate a corresponding global scene description. The description is then passed into an off-the-shelf image generator to create a meaningful global scene image. Finally, this image is further deconstructed by the VLM into multiple local scenes and story sub-prompts, serving as the basis for subsequent story generation. This scene planning strategy ensures scene-level narrative continuity, as the local scenes are derived from a global scene with shared scene semantics.

(ii) To maintain *scene consistency*, we design a novel **Long-Term Scene-Sharing Attention** mechanism. Specifically, it first employs a *Mask-Guided Scene Injection* module, which enhances the IP-Adapter [20] with cross-attention masks to guide fine-grained scene injection, thereby ensuring subject style diversity. Then, the latent representations interact across scenes through a *Scene-Sharing Attention* module during the denoising process, thereby preserving scene consistency across generated stories. Furthermore, this attention module is further extended by an *Extrapolable Noise Blending* scheme, thereby achieving long-term scene consistency across stories with low overhead.

Extensive qualitative and quantitative comparisons validate the effectiveness of our SceneDecorator, with ablation studies and diverse applications showcasing its robustness and versatility.

# 2 Related Works

**Controllable Text-to-Image Generation.** Given the ambiguity of textual descriptions in guiding image style [21, 8], content [22, 23], and layout [24], many works have been dedicated to enhancing control in text-to-image (T2I) generation. Prior works like ControlNet [25] and T2I-Adapter [26] tackle this challenge by employing trainable modules. These modules enhance control over visual style and spatial organization, making them more effective than naive T2I models [27, 28, 29, 30]. In addition, some studies have also explored several advanced techniques like prompt engineering [31] and cross-attention constraints [32, 33], enabling better generation regulation. Moreover, some approaches focus on visual content generation with diverse task paradigms [34, 35, 36, 37, 38], while others focus on more practical generation in real-world applications [39, 40, 41, 42, 43].

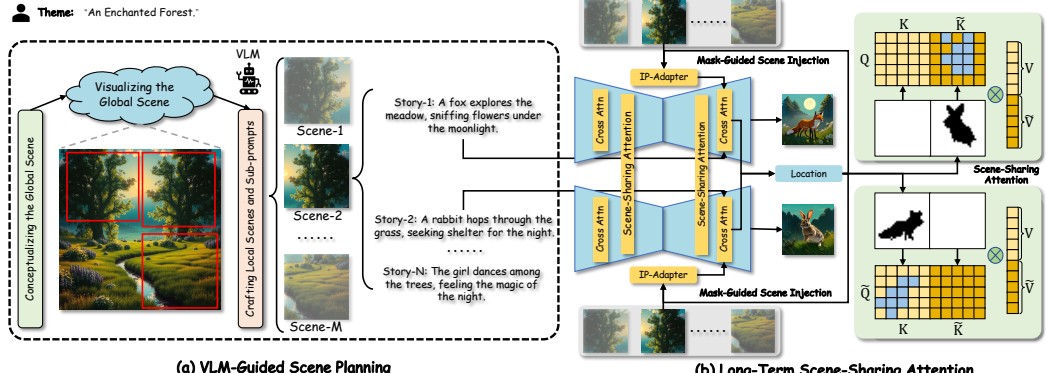

Figure 2: Overall framework of SceneDecorator. (a) VLM-Guided Scene Planning involves conceptualizing, visualizing, and crafting in a "global-to-local" manner. (b) Long-Term Scene-Sharing Attention maintains long-range scene consistency and subject diversity across generated stories.

**Story Generation.** Owing to the success of diffusion models, many recent works have applied them to story generation, showcasing significant value in real-world applications. Initially, AR-LDM [44] uses an auto-regressive paradigm for story generation, while Make-A-Story [45] integrates a visual memory module for aggregation. Subsequently, some researchers attempt to leverage large language models (LLMs) for coherent story generation. For example, StoryGPT-V [46] uses LLMs to resolve ambiguous references and maintain context, while SEED-Story [47] combines image-text data to generate coherent story images. Recently, some works [48, 49, 50] have begun to focus on character consistency in story generation, ensuring that their identity remains intact across diverse text descriptions. Building on this foundation, other studies [5, 14] manipulate attention maps to achieve training-free story generation while maintaining character consistency. However, these methods overlook the planning and consistency of scene contexts in story generation, which also play a fundamental role in visual storytelling. Our work seeks to resolve these problems systematically.

## 3 Methodology

### 3.1 Overall Pipeline

In this work, we design a training-free framework called **SceneDecorator**, to address two key challenges in story generation: *scene planning* and *scene consistency*. The overall framework of SceneDecorator is illustrated in Figure 2, which comprises two core techniques: (i) *VLM-Guided Scene Planning.* Leveraging a powerful Vision-Language Model (VLM) as a director, it decomposes user-provided themes into local scenes and story sub-prompts in a "global-to-local" manner. (ii) *Long-Term Scene-Sharing Attention.* By simultaneously integrating mask-guided scene injection, scene-sharing attention, and extrapolable noise blending, it maintains subject style diversity and long-term scene consistency in story generation. We elaborate on these in the following sections.

### 3.2 VLM-Guided Scene Planning

Relying solely on text descriptions to generate story images often lacks scene-level narrative coherence. In real-world applications like filmmaking, a global scene is first established, and then local scenes are derived to unfold different narratives. Inspired by this application, we propose VLM-Guided Scene Planning, which leverages the visual understanding of VLM to unfold scene shots and related narratives in a "global-to-local" manner. The overall process is decomposed into three core steps: (i) *Conceptualizing the Global Scene*, (ii) *Visualizing the Global Scene*, and (iii) *Crafting Local Scenes and Sub-prompts*. In the following, we elaborate on each stage in detail.

**Conceptualizing the Global Scene.** We first leverage a powerful VLM to provide a comprehensive global scene description. Specifically, when users provide a theme $\mathcal{T}$, we expect the VLM $\mathcal{F}_\theta$ to fully exploit its scene imagination capability and generate a global scene description $Q = \mathcal{F}_\theta(\mathcal{T})$ related to the given theme $\mathcal{T}$. Moreover, we further enhance VLM performance by leveraging its

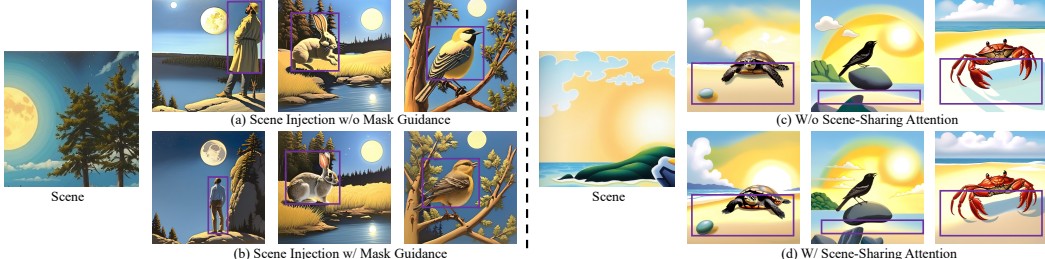

Figure 3: Comparison of different methods. In (a), subject styles align with the scene but at the expense of diversity, whereas (b) better showcases diversity. Compared to (c), (d) further emphasizes scene consistency. Note that *purple boxes* highlight distinctions. *Best viewed with zoom-in.*

in-context learning ability. Specifically, we provide illustrative examples to guide the model toward more accurate outputs, with the example details presented in the supplemental material.

**Visualizing the Global Scene.** Based on the global scene description $Q$ produced by the VLM $\mathcal{F}_\theta$ above, we then employ a powerful off-the-shelf T2I model, like FLUX.1-dev [51], to generate a meaningful global scene image $\mathcal{V}$. In summary, using the complementary capabilities of the VLM and the T2I model, we transform the abstract theme $\mathcal{T}$ into an immersive global scene image $\mathcal{V}$. This scene image establishes the global foundation for subsequent local storyline creation.

**Crafting Local Scenes and Sub-prompts.** Building upon the generated global scene image $\mathcal{V}$ above, we further leverage the powerful perception capability of the VLM to intricately create local storylines, which contain relevant local scenes and story sub-prompts. Specifically, we expect the VLM $\mathcal{F}_\theta$ to act as an imaginative director: based on the user-provided theme $\mathcal{T}$, it determines the coordinates $\{\mathcal{L}^i\}_{i=1}^M$ for $M$ storyboard scenes within the global scene image $\mathcal{V}$. Then, the global scene image is cropped accordingly to extract final local scenes $\{\mathcal{V}^i\}_{i=1}^M$. This procedure is formulated below:

$$\{\mathcal{V}^i\}_{i=1}^M = \text{Crop}(\mathcal{V}, \{\mathcal{L}^i\}_{i=1}^M), \quad \text{where } \{\mathcal{L}^i\}_{i=1}^M = \mathcal{F}_\theta(\mathcal{V}, \mathcal{T}). \tag{1}$$

Finally, for each cropped local scene $\mathcal{V}^i$ and the corresponding theme $\mathcal{T}$, we employ the VLM $\mathcal{F}_\theta$ to generate $N$ sequential story sub-prompts $\mathcal{P}^{1:N}$. This process can be formulated as follows:

$$\mathcal{P}^{1:N} = \mathcal{F}_\theta(\mathcal{V}^i, \mathcal{T}), \quad i = 1, \cdots, M. \tag{2}$$

This scene planning framework transforms the user-provided abstract theme $\mathcal{T}$ into multiple local scenes and corresponding story sub-prompts, creating a cohesive narrative that enhances storytelling. The detailed prompts used by the VLM in each step are provided in the supplemental material.

### 3.3 Long-Term Scene-Sharing Attention

Once each local scene $\{\mathcal{V}^i\}_{i=1}^M$ is established, it is typically combined with corresponding story sub-prompts $\mathcal{P}^{1:N}$ for subsequent story generation, which is similar to film storyboarding [19]. During generation, we propose Long-Term Scene-Sharing Attention to address the challenge of scene consistency that is overlooked in prior work. First, *Mask-Guided Scene Injection* is developed to preserve the diversity of subject style while achieving scene injection. Next, *Scene-Sharing Attention* is utilized to maintain scene consistency across multiple stories. Furthermore, this attention mechanism is further extended through *Extrapolable Noise Blending* to achieve long-term consistency with low memory overhead. The details of these components are described in the following paragraphs.

**Mask-Guided Scene Injection.** Achieving scene consistency first requires the effective injection of visual semantics from the given scene. One straightforward approach is IP-Adapter [20], which enhances representation by integrating visual and text prompt through a decoupled cross-attention mechanism. However, as shown in Figure 3(a), direct using IP-Adapter for scene injection preserves overall semantics but makes the subjects blend too tightly with the background, which reduces the style diversity across generated stories. To address this issue, we improve IP-Adapter with cross-attention masks to guide fine-grained scene injection, thereby ensuring subject style diversity.

During the cross-attention process, local scene $\mathcal{V}^i$ and story sub-prompt $P^j$ are first encoded into hidden features and then mapped to $K'_c, V'_c$ and $K_c, V_c$ via respective weight matrices. Next, the

---
**Algorithm 1** Extrapolable Noise Blending
---

**Input:** $T_1, T_2$                                                The time interval of noise blending
**Input:** $N$                                                     The numbers of generated stories
**Input:** $\mathcal{P}^{1:N}$                     The text descriptions of different stories
**Input:** $\mathcal{V}$                                                      The visual prompt of scene
**Input:** $\varepsilon_\theta$, DDIMSchedule                    Diffusion model, noise scheduling
**Output:** $\{I^i\}_{i=1}^N$                    Different stories generated by the model

**for** $t = T, T-1, \ldots, 3, 2$ **do**
     $\varepsilon_{tmp}^{1:N} \leftarrow 0$
     **if** $t \geq T_1$ and $t \leq T_2$ **then**
         $\mathcal{S} \leftarrow \{(i,j)|i,j \in 1, \ldots, N, i \neq j\}$
         $norm \leftarrow N - 1$
         **for** $(i,j) \in \mathcal{S}$ **do**
              $\varepsilon_1, \varepsilon_2 \leftarrow \varepsilon_\theta^{i,j}(Z_t^{i,j}, t, \mathcal{P}^{i,j}, \mathcal{V})$       *# Mask-Guided Scene Injection and Scene-Sharing Attention*
             $\varepsilon_{tmp}^i \leftarrow \varepsilon_{tmp}^i + \varepsilon_1$
             $\varepsilon_{tmp}^j \leftarrow \varepsilon_{tmp}^j + \varepsilon_2$
         **end for**
     **else**
         $norm \leftarrow 1$
         **for** $k = 1, 2, \ldots, N-1, N$ **do**
             $\varepsilon_{tmp}^k \leftarrow \varepsilon_{tmp}^k + \varepsilon_\theta^k(Z_t^k, t, \mathcal{P}^k, \mathcal{V})$                          *# General denoising*
         **end for**
     **end if**
     $Z_{t-1}^{1:N} \leftarrow$ DDIMSchedule$(\varepsilon_{tmp}^{1:N}/norm, Z_t^{1:N}, t)$                          *# Blend the noises*
**end for**
$I^{1:N} \leftarrow \mathcal{D}(Z_1^{1:N})$
**Return:** $\{I^i\}_{i=1}^N$

---

latent representation is mapped to $Q_c$ and multiplied by $K_c$ and $K_c'$ to generate two attention maps:

$$\mathcal{A}_c = \mathrm{Softmax}\left(\frac{Q_c \cdot K_c^T}{\sqrt{d}}\right), \quad \mathcal{A}_c' = \mathrm{Softmax}\left(\frac{Q_c \cdot K_c'^T}{\sqrt{d}}\right), \tag{3}$$

where $d$ is the dimension of $Q_c$ and $K_c$. $\mathcal{A}_c \in \mathbb{R}^{(hw) \times L}$ represents the cross-attention map between text and generated image, where $hw$ denotes the number of image tokens and $L$ indicates the number of text tokens. $\mathcal{A}_c'$ denotes the cross-attention map between scene and generated image.

At each denoising step, the cross-attention map $\mathcal{A}_c$ is averaged over all previous steps. The subject token of the sub-prompt $P^j$ is then selected, and its activation region in $\mathcal{A}_c'$ is used as the masks $\mathcal{M} \in \mathbb{R}^{h \times w}$. Finally, the cross-attention maps $\mathcal{A}_c$ and $\mathcal{A}_c'$ are multiplied by $V_c$ and $V_c'$, respectively, and the results are combined through an element-wise weighted sum with the subject masks $\mathcal{M}$:

$$Z_c^{\mathrm{new}} = \mathcal{A}_c \cdot V_c + \lambda \cdot (1 - \mathcal{M}) \cdot \mathcal{A}_c' \cdot V_c', \tag{4}$$

where $\lambda$ is a weighting factor that balances scene features and text features. As shown in Figure 3(b), this approach ensures effective scene injection while enhancing the diversity of subject styles.

**Scene-Sharing Attention.** The above cross-attention mechanism effectively injects scene semantics, achieving scene consistency across generated stories to some extent. However, as shown in Figure 3(c), there is an inherent conflict between story sub-prompts and scene consistency, which significantly weakens coherence across generated stories. To resolve this issue, we extend self-attention with scene-sharing attention to further enhance scene consistency between generated stories.

During the self-attention process, the latent representations from dual-branches are mapped to $Q, K, V$ and $\tilde{Q}, \tilde{K}, \tilde{V}$ through their weight matrices. As depicted in Figure 2(b), each branch then attends to the $\tilde{K}$ and $\tilde{V}$ of the other branch for scene interaction, with the masks $\tilde{M}$ applied to restrict attention to the background. The new key $K'$ and new value $V'$ are formulated as follows:

$$K' = [K, \tilde{K} \odot (1 - \tilde{M})], \quad V' = [V, \tilde{V} \odot (1 - \tilde{M})], \tag{5}$$

where $[*]$ represents the concatenation operation and $\odot$ denotes element-wise product operation. It is noted that the subject masks $\mathcal{M}$ for the other branch are derived using the same method outlined in *Mask-Guided Scene Injection* and are therefore omitted here for brevity.

Table 1: Quantitative comparison of automatic metrics and user study across other baselines. The best result is marked in **bold**, and the second-best is underlined.

| Methods | Automatic Metrics | | | User Study | | |
|---|---|---|---|---|---|---|
| | CLIP-T ↑ | DreamSim-I ↓ | DINO-F ↑ | Text Align. ↑ | Scene Align. ↑ | Image Qual. ↑ |
| CustomDiffusion [52] | 0.306 | 0.752 | 0.373 | 7.9% | 3.4% | 6.0% |
| ConsiStory [5] | **0.320** | 0.723 | 0.475 | 21.3% | 14.1% | 24.7% |
| StoryDiffusion [14] | 0.311 | 0.735 | 0.340 | 14.3% | 6.3% | 11.8% |
| SceneDecorator (Ours) | 0.312 | **0.605** | **0.571** | **56.5%** | **76.2%** | **57.5%** |

Finally, the $Q$ and $\tilde{Q}$ from each branch will perform attention with the new $K'$ and $V'$ respectively:

$$\text{Attention}(Q, K', V'), \quad \text{Attention}(\tilde{Q}, K', V'). \tag{6}$$

As shown in Figure 3(d), this mechanism allows different stories to attend each other's scene information during the self-attention process, further enhancing scene consistency across stories.

**Extrapolable Noise Blending.** Although the above method ensures scene consistency across stories, it is limited to generating two stories. We propose an extrapolable noise blending scheme, achieving long-term scene consistency across multiple stories with low overhead, as shown in Algorithm 1.

To simultaneously generate $N$ stories with consistent scenes, we extend the *Scene-Sharing Attention* module with noise blending during the denoising interval $t \in [T_1, T_2]$. Specifically, the latent representations $\{Z_t^i\}_{i=1}^N$ are dynamically partitioned into complementary pairs $< Z_t^i, Z_t^j >$, with $i, j \in 1, ..., N$, allowing each story to participate in $N - 1$ pairings per denoising step. The noise predicted for each story in different pairs is then averaged to further update the latent representations. This noise blending strategy enables scene interaction across multiple stories while requiring the GPU memory usage of only two stories, therefore ensuring significantly lower overhead.

## 4 Experiments

### 4.1 Experimental Setups

**Implementation Details.** We leverage Qwen2-VL [53] as the VLM to guide scene planning, FLUX.1-dev [51] as the off-the-shelf T2I model to generate global scenes, and SDXL [54] as the base model to collaborate with the proposed techniques for story generation. Additionally, we employ IP-Adapter-XL [20] to support extra scene input. The hyperparameters are set as follows: $M = 4$, $N = 5$, $T_1 = 0$, and $T_2 = 25$. SceneDecorator can run on a single RTX 3090 GPU without further training.

**Baselines and Datasets.** Since our work is the first to focus on scene-oriented story generation, there are no directly related comparison methods. Therefore, we select and adapt three baselines: CustomDiffusion [52], ConsiStory [5], and StoryDiffusion [14], due to their competitive performance on similar tasks. To validate the effectiveness of SceneDecorator, we use GPT-4o [55] to randomly generate 146 themes across different domains. Each theme is then decomposed into 4 distinct local scenes by the VLM, with each scene containing 5 story sub-prompts. In total, there are 2,920 scene-prompt pairs that serve as input for each method, ensuring a fair comparison.

### 4.2 Quantitative Comparisons

We validate the superiority of our SceneDecorator from two perspectives: *Automatic Metrics* that provide an objective assessment, and *User Study* that offers a subjective evaluation.

**Automatic Metrics.** We evaluate the quality of the generated stories from three dimensions: *(i) CLIP-T* [56], which assesses the alignment between the generated stories and the input prompt, *(ii) DreamSim-I* [57], which measures the alignment between the generated stories and the input scene, and *(iii) DINO-F* [58], which evaluates the scene consistency across generated stories. The detailed results are reported in Table 1. SceneDecorator outperforms all other methods in both the DreamSim-I and DINO-F metrics, with ConsiStory [5] ranking second. This indicates that our method achieves the best performance in scene alignment and consistency. In the CLIP-T metric, our SceneDecorator ranks

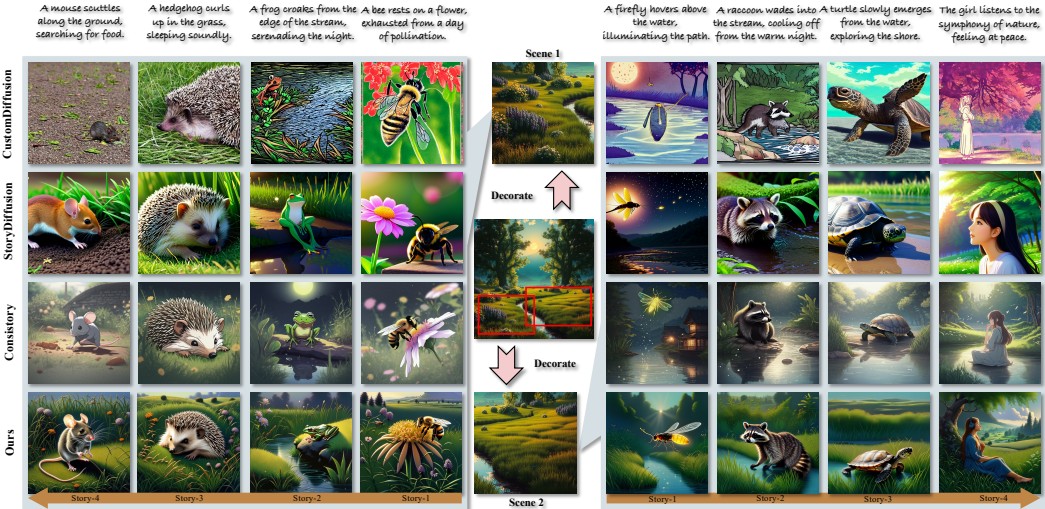

Figure 4: Qualitative comparison of our SceneDecorator with other baselines. SceneDecorator demonstrates superior scene consistency and alignment across different stories compared to other baselines, making it well-suited for creative applications in filmmaking. *Best viewed with zoom-in*.

second, slightly behind ConsiStory. Overall, SceneDecorator demonstrates superior performance across these metrics, showing its effectiveness in scene-oriented story generation.

**User Study.** We designed a questionnaire with 13 groups of generated results, where each group contains four different stories. Questionnaires are randomly distributed to participants from diverse countries, cultural backgrounds, genders, and age groups, inviting them to select the best result from each group based on three key aspects: *text alignment*, *scene alignment*, and *image quality*. Ultimately, we have received 61 valid responses, with the detailed results illustrated in Table 1. Our SceneDecorator achieves state-of-the-art performance across all three aspects, demonstrating particularly significant gaps in scene alignment. This success can be attributed to our innovative VLM-Guided Scene Planning strategy and the advanced Long-Term Scene-Sharing Attention mechanism.

### 4.3 Qualitative Comparisons

In addition, we conduct qualitative comparisons of the proposed SceneDecorator with three existing approaches, including CustomDiffusion [52], StoryDiffusion [14], and ConsiStory [5]. The visualization results are presented in Figure 4, and additional results can be found in the supplementary material. CustomDiffusion, which is designed for personalized characters, faces challenges in generating personalized scenes. Similarly, StoryDiffusion, which is focused on consistent character story generation, struggles to maintain scene consistency across different stories. On the other hand, Consistory demonstrates strong performance in preserving scene consistency but encounters difficulties in effectively capturing the full scope of scene information, limiting its versatility. In contrast, our SceneDecorator efficiently capture detailed semantics of the scene while ensuring scene consistency across generated stories, showcasing its superiority in scene-oriented story generation.

### 4.4 Ablation Studies

In this section, we explore the effectiveness of the three proposed components: *Mask-Guided Scene Injection*, *Scene-Sharing Attention*, and *Extrapolable Noise Blending*, individually.

**Mask-Guided Scene Injection.** We compare the generation results with and without the mask-guided scene injection module, with qualitative examples shown in Figure 5. Without mask-guided scene injection, textual descriptions alone fail to generate stories that contain specific scene semantics. In contrast, incorporating mask-guided scene injection provides fine-grained guidance, which not only facilitates the injection of scene semantics but also preserves diversity in subject styles.

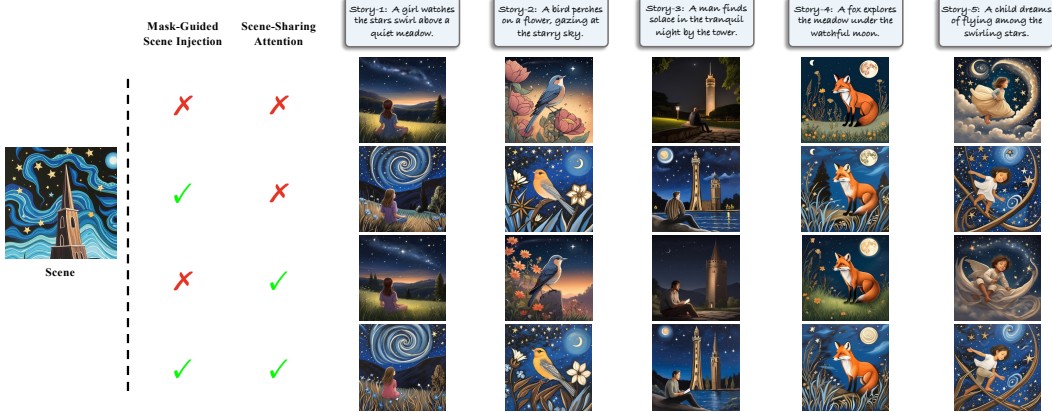

Figure 5: Ablation study of the two components: *Mask-Guided Scene Injection* and *Scene-Sharing Attention*. "✓" and "✗" indicate whether each component is used. The synergy between these components ensures scene consistency and subject diversity across generated stories.

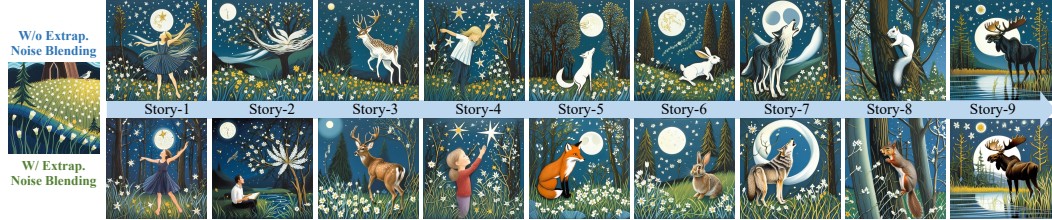

Figure 6: Comparison of long-term consistency with and without *Extrapolable Noise Blending*. Our approach ensures consistent scenes with low overhead, while effectively preserving subject diversity throughout the entire narrative, leading to more cohesive results. *Best viewed with zoom-in*.

Table 2: Efficiency analysis of *Extrapolable Noise Blending*. "OOM" represents out of memory.

| Methods | Number of Generated Story Images | | | | | | |
|---|---|---|---|---|---|---|---|
| | 1 | 2 | 5 | 10 | 15 | 20 | 25 |
| w/o Extrap. Noise Blending | 11.4G | 12.7G | 14.5G | 17.5G | 20.4G | 23.5G | OOM |
| w/ Extrap. Noise Blending | 11.4G | 12.7G | 12.7G | 12.7G | 12.7G | 12.7G | 12.7G |

**Scene-Sharing Attention.** Moreover, the effectiveness of the scene-sharing attention mechanism is also demonstrated in Figure,5, where its impact on maintaining narrative coherence is clearly illustrated. By leveraging this proposed attention mechanism, the model is able to enhance scene consistency across different generated stories, ensuring that key contextual elements remain fully aligned and logically connected throughout the narrative. When combined with the scene injection strategy, this synergy not only ensures that subject diversity is effectively integrated into the storytelling process but also reinforces scene consistency across different generated stories.

**Extrapolable Noise Blending.** Finally, we assess the effectiveness of extrapolable noise blending in generating stories that preserve long-term scene consistency, with the associated experiments performed on a single RTX 3090 GPU. The visualization result is illustrated in Figure 6 and the memory usage is reported in Table 2. Although scene consistency can be partially maintained without extrapolable noise blending, the associated memory usage scales with the number of story images and often causes out-of-memory (OOM) errors. In contrast, applying extrapolable noise blending fixes memory usage, effectively preventing "OOM" issues while ensuring long-term scene consistency.

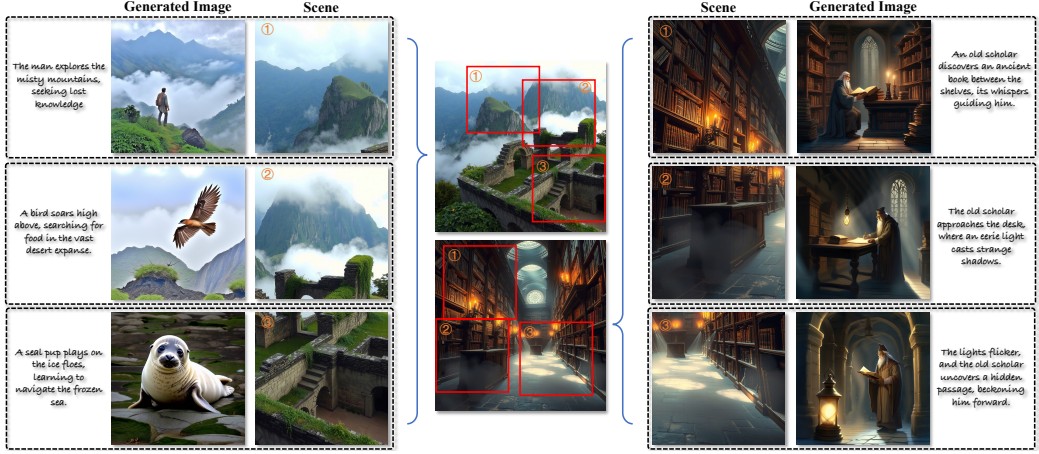

Figure 7: More applications of our SceneDecorator. It can support generation with manual scene input (*left*) and generation with consistent characters (*right*). *Best viewed with zoom-in*.

## 4.5 Rationality Analysis of VLM-Guided Scene Planning

We further analyze the local scenes partitioned by the VLM-Guided Scene Planning strategy from two perspectives: *Coordinate Rationality* and *Semantic Rationality*, as detailed below.

**Coordinate Rationality.** When partitioning the global scene image (Equation 1), the coordinates predicted by the VLM may occasionally fall slightly outside the defined image boundaries. To robustly address this issue, we apply a simple yet effective correction: invalid coordinates are automatically snapped to the nearest valid bounding box, thereby ensuring all generated coordinates remain usable.

**Semantic Rationality.** To assess the semantic rationality of the partitioned local scenes, we conducted a quantitative evaluation using GPT-4o [55]. For each sample, the story theme, the global scene, the derived local scenes, and the corresponding story sub-prompts are provided to GPT-4o, which is instructed to evaluate them along three key criteria: *Narrative Coherence*, *Theme Adherence*, and *Layout Reasonableness*. Each criterion was scored on a 10-level scale (0–100%), with the details illustrated in Table 3. The results show that VLM-Guided Scene Planning exhibits strong robustness across narrative coherence, theme adherence, and layout reasonableness.

Table 3: GPT-4o evaluation of narrative coherence, theme adherence, and layout reasonableness.

| Narrative Coherence | Theme Adherence | Layout Reasonableness |
|---|---|---|
| 90.06% | 92.57% | 90.29% |

## 5 More Applications

**Generation with Manual Scene Input.** In addition to automatic VLM-Guided Scene Planning, our SceneDecorator can also support manual scene input. Specifically, users can provide a global scene, manually divide it into local scenes, and use them for subsequent story generation. As illustrated in Figure 7, the impressive visual results further emphasize the scalability of SceneDecorator.

**Generation with Consistent Character.** Beyond generating stories with scene consistency, SceneDecorator can also generate stories that preserve character consistency under different scenes. Specifically, we modify the scene-sharing attention by inverting the mask to ensure character consistency, while keeping the mask-guided scene injection unchanged. In Figure 7, the same character experiences different stories across scenes, further showcasing the flexibility of SceneDecorator.

**Generation with Other Tools.** As a training-free framework, SceneDecorator can also seamlessly integrate with other generative tools to meet diverse user needs. Detailed examples are demonstrated in Figure 8. It can be combined with PhotoMaker [59] for customized character generation and with ControlNet [25] for precise conditional control. Furthermore, it can work effectively with stylized LoRAs [60] to achieve diverse style generation. In summary, by incorporating diverse generative tools, our proposed SceneDecorator highlights more creative and flexible workflows.

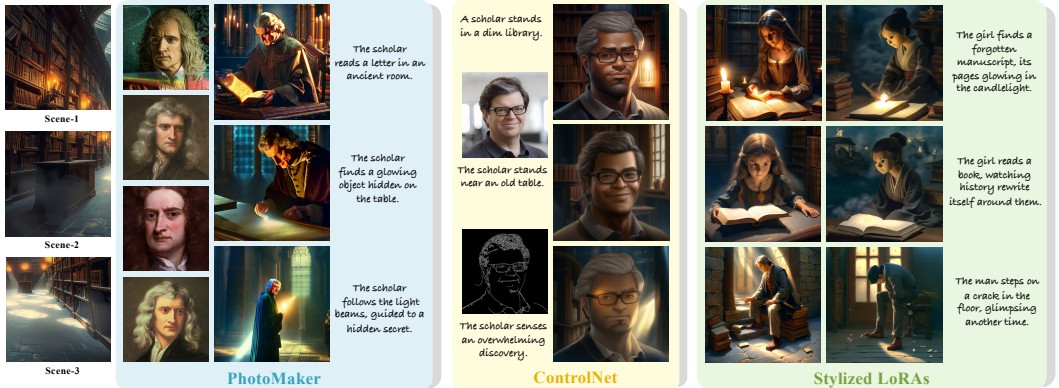

Figure 8: More applications of our SceneDecorator. It can also support generation with other tools: PhotoMaker, ControlNet, and stylized LoRAs. *Best viewed with zoom-in*.

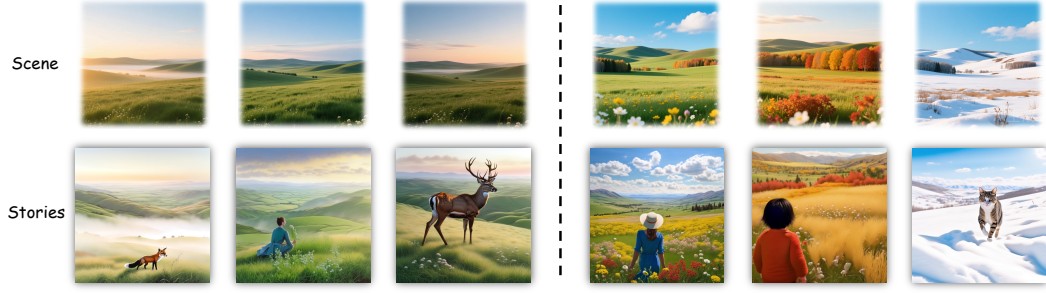

Figure 9: More applications of our SceneDecorator. It can also support generation with evolving scenes, such as transitions from morning to dusk (*left*) as well as from summer to winter (*right*).

**Generation with Evolving Scenes.** In addition, SceneDecorator provides robust support for generating multiple stories with evolving scenes, such as changes in the time of day or shifts in seasons, which further showcases its flexibility and adaptability. As illustrated in Figure 9, the model is capable of dynamically adapting to a wide range of scene inputs, enabling the generation of diverse stories that evolve seamlessly across different contexts. SceneDecorator empowers users to flexibly craft immersive and dynamic stories that capture the essence of change across diverse settings.

## 6 Conclusion

This paper introduces **SceneDecorator**, a training-free framework for *scene-oriented story generation*. It emphasizes scene planning and scene consistency, in contrast to the character consistency focus of prior works. Our SceneDecorator comprises two core techniques: (i) *VLM-Guided Scene Planning*, which decomposes user-provided themes into local scenes and story sub-prompts in a "global-to-local" manner, and (ii) *Long-Term Scene-Sharing Attention*, which integrates mask-guided scene injection, scene-sharing attention, and extrapolable noise blending to maintain subject style diversity and long-term scene consistency during generation. Extensive experiments validate the effectiveness of SceneDecorator, showcasing its ability to enhance creativity across real-world applications.

**Acknowledgments.** This work was supported by the InnoHK initiative of the Innovation and Technology Commission of the Hong Kong Special Administrative Region Government via the Hong Kong Centre for Logistics Robotics, by the Faculty Initiatives Research of Monash University (Contract No. 2901912), by the NVIDIA Academic Hardware Grant Program, and by the Research Start-up Fund for Prof. Xiaowei Hu at the Guangzhou International Campus, South China University of Technology (Grant No. K3250310).

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

# A Implementation Details about Baselines

**CustomDiffusion** is a tuning-based method that requires fine-tuning to inject given visual concepts. Specifically, we fine-tune it using the local scene images obtained from our VLM-guided scene planning framework. During inference, we adopt the corresponding story sub-prompts together with the fine-tuned model to enable a fair comparison.

**StoryDiffusion** is a training-free method designed for subject-consistent story generation and naturally supports image conditioning. We directly use the local scene images and the associated story sub-prompts from the VLM-guided scene planning framework as input.

**ConsiStory** is also a training-free method for subject-consistent story generation. However, the official implementation does not support images as input. For a meaningful comparison, we make minimal modifications to adapt it to our task, prepending the local scene images to the input batch and leveraging it as a reference through self-attention and feature infusion mechanisms. Additionally, the same story sub-prompts are also adopted accordingly.

# B Implementation Details about VLM-Guided Scene Planning

We propose VLM-Guided Scene Planning to guide the VLM in decomposing the user-provided theme into multiple local scenes and stories with specific instructions. The VLM is leveraged in two step, including *conceptualizing the global scene* and *crafting local scenes and sub-prompts*.

**Conceptualizing the Global Scene.** The complete prompt for conceptualizing the global scene consists of two main components, which are also detailed in Table 4.

- *Instruction:* This specifies the task and defines the specific theme from the user prompt, helping the VLM perform effectively in global scene generation tasks. It provides a structured way to guide the model in synthesizing scenes and storylines.

- *Examples:* This is an example that shows how the VLM works. The user prompt is filled into the provided system instruction. The model then continues the conversation based on the user prompt and provides the scene description in the specified format.

Table 4: Complete prompt for conceptualizing the global scene.

| Role | Content |
|---|---|
| Instruction | **System**: "You are now a scene planner tasked with using your rich imagination to design a global scene based on the theme provided by the user. Your job is to describe the environment and layout of the scene as thoroughly as possible and then output it into a concise description of no more than 50 words. This description will be used to generate the corresponding global scene, which can include various styles such as anime, realism, art, abstract, etc. Note: You only need to output the final summary result, and it must not include any theme or character information." Theme: "{User Prompt}." Please provide the corresponding output." |
| Examples | **User**: "Snowy dreams and falling stars." 
 **VLM**: [A misty forest at dawn, bathed in soft golden light filtering through ancient trees. Delicate ferns and moss-covered rocks line winding paths, while a serene stream meanders through, reflecting the sky's pale hues. Birdsong fills the air, and gentle breezes stir the leaves, creating a peaceful, dreamlike atmosphere.] |

**Crafting Local Scenes and Sub-prompts.** Complete prompt for crafting local scenes also includes two main components, which are also detailed in Table 5.

- *Instruction:* The model is required to select four distinct local scenes (sub-scenes) from an image, ensuring minimal overlap, and providing the coordinates for each local scene. Subsequently, we mandate the creation of five unique stories for each local scene, focusing on a protagonist's experience, summarized in one sentence. The protagonist can be an animal or a human with generic pronouns. The instruction emphasizes creativity and coherence,

ensuring captivating narratives while maintaining uniqueness. The output should include the coordinates of local scenes and the corresponding story sub-prompts for local scenes.

- *Examples:* There is also an example that shows how to craft local scenes. The user needs to provide the generated global image, the VLM then continues the conversation based on the user prompt and provides the sub-scene description as required.

Table 5: Complete prompt for crafting local scenes and stories.

| Role | Content |
|------|---------|
| Instruction | **System**: " Now you are a story director. Please fully utilize this 1024x1024 image, unleash your imagination, and design suitable scenes from a spatial perspective for the theme "" while creating corresponding storylines for each scene from a temporal perspective. Spatial Design: Select 4 distinct sub-scenes with minimal overlap. Provide top-left and bottom-right coordinates in the format: [x1, y1, x2, y2]. Temporal Storytelling: Create 5 unique stories for each sub-scene (20 stories in total), focusing on a protagonist's experience in each scene. Story Summary: Each story must reflect the protagonist's action or experience and be summarized in one sentence (max 15 words). Protagonist Setup: The protagonist can be an animal or a generic human (e.g., "girl," "man"). Creativity and Coherence: Ensure creativity and engagement while maintaining uniqueness and narrative coherence. Output Format: Provide coordinates in [x1, y1, x2, y2] format for each sub-scene, followed by the stories for each. " |
| Examples | **User**: "[The global image presented in Figure 4 of the main paper.]"
**VLM**: "[Location of a local scene]: [18, 8, 506, 499]. 1.A fox explores the meadow, sniffing flowers under the moonlight. 2.The girl dances among the trees, feeling the magic of the night. 3.An owl perches on a branch, watching over the serene landscape. 4.A rabbit hops through the grass, seeking shelter for the night. 5.A deer grazes quietly, enjoying the peaceful evening ......" |

# C   Limitations and Future Work

Our SceneDecorator is tailored for scene planning and ensuring consistency, showcasing notable advantages compared to existing methods. Nonetheless, several limitations remain: (i) As a training-free method, the story generation capability of SceneDecorator largely depends on the underlying foundation models, such as FLUX.1, SDXL, and Qwen2-VL. Consequently, any limitations inherent in these base models can constrain the overall performance of SceneDecorator. (ii) For scene injection, we adopt the IP-Adapter technique, which proves effective in general. However, it performs less reliably in out-of-distribution scenarios, such as depicting an elephant in the sky.

Regarding the first limitation, future work could explore more complex scene-oriented story generation tasks, such as scene transitions and multi-scene integration. As for the second limitation, developing more effective scene control mechanisms beyond the current reliance on IP-Adapter would be a promising direction.

# D   Potential Negative Societal Impact

Our work is primarily designed for scene-oriented story generation within the broader domains of visual content creation. However, we explicitly acknowledge that technologies capable of generating multi-image narratives may also pose significant societal risks if misused. In particular:

- **Disinformation and Propaganda.** The ability to generate visual narratives could be exploited to fabricate persuasive but false stories, amplifying the spread of disinformation.

- **Bias and Stereotypes.** Unintended biases present in the input story themes could potentially reinforce harmful cultural stereotypes or discriminatory visual representations.

- **Inappropriate or harmful content.** Without proper safeguards and regulatory oversight, the generated content might unintentionally include sensitive, violent, or inappropriate material, potentially causing psychological or emotional harm to diverse audiences.

We highlight these important concerns to encourage the responsible and ethical use of our method and emphasize the importance of developing safeguards against potential misuse.

# E    Additional Results

To further validate the effectiveness and versatility of SceneDecorator, we present additional qualitative comparisons in Figure 10 and Figure 11. These examples highlight the model's ability to generate coherent and contextually appropriate scenes across diverse prompts and settings.

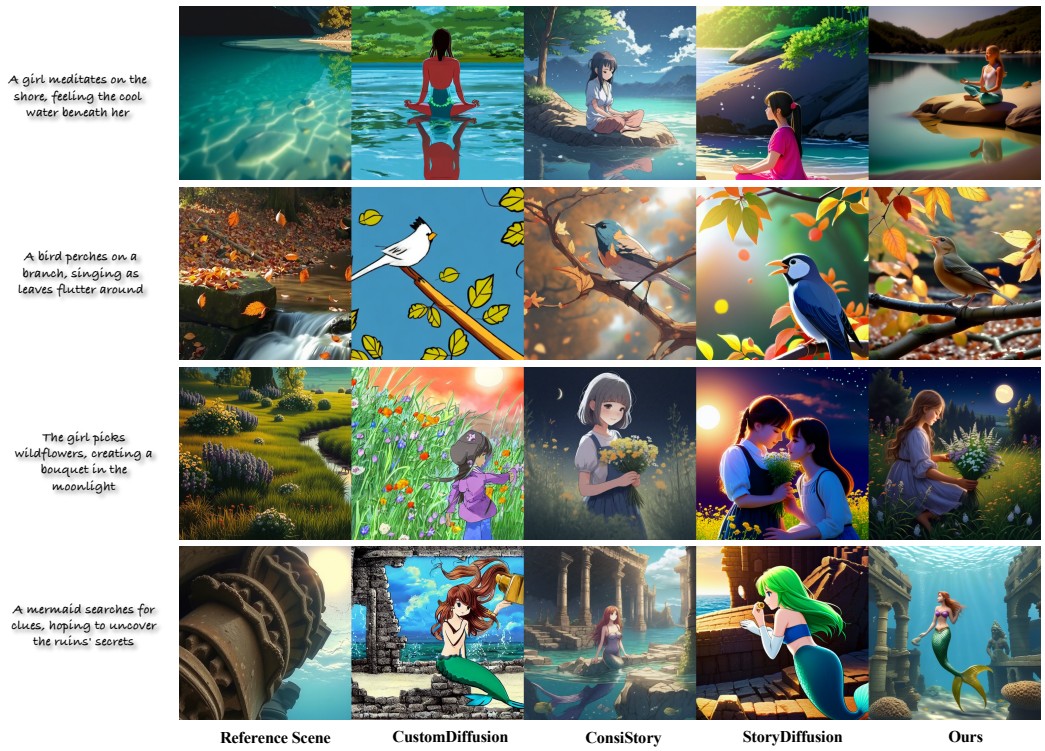

Figure 10: Additional qualitative comparisons. Our method effectively follows the text prompt while maintaining scene alignment.

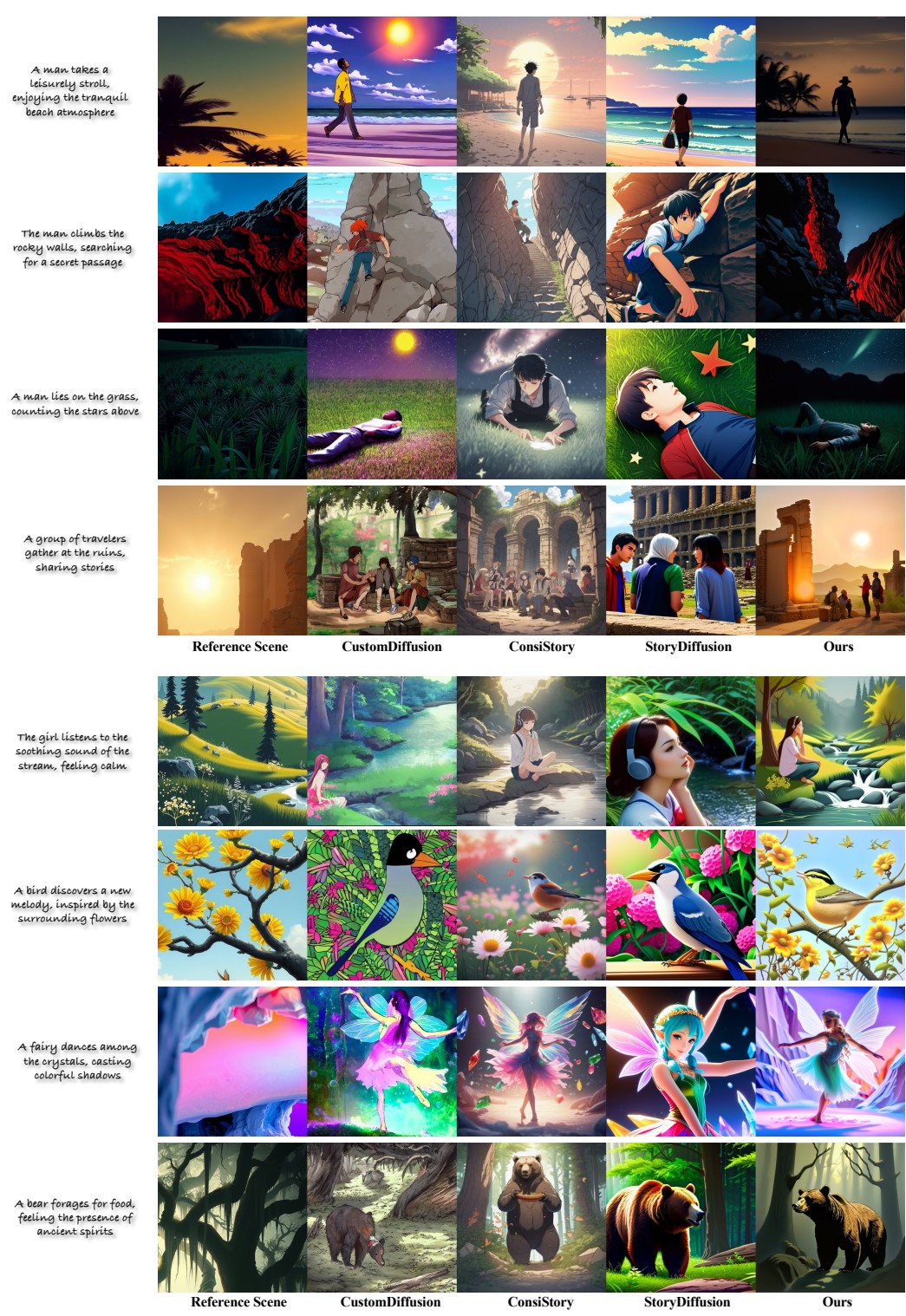

Figure 11: Additional qualitative comparisons. Our method successfully follows the prompt while maintaining scene alignment.

