# OpenReview forum: "SceneDecorator: Towards Scene-Oriented Story Generation with Scene Planning and Scene Consistency"
_NeurIPS.cc/2025/Conference — NeurIPS 2025 poster_

### Official Review · Reviewer_ohEf · 2025-06-12

**Clarity:** 2
**Significance:** 2
**Originality:** 2
**Rating:** 3
**Confidence:** 5

**Summary:**

This paper proposes SceneDecorator, which can decorate a patch of the input image with story prompts generated by VLM and output the decorated images with additional contents. The method leverages diffusion models with masks for the generation. The experiment has qualitative and quantitative results, which try to prove the effectiveness of this proposed method.

**Questions:**

If the authors could answer my question in the weaknesses section, and solve the generation consistency problem, and ensure that it's truly a decoration, not a re-creation, I would raise my points to positive.

**Ethical Concerns:**

["NO or VERY MINOR ethics concerns only"]

**Final Justification:**

After discussion with the author, I believe that SceneDecorator failed to generate a consistent story with a complete storyline, although the authors claim so in the rebuttal and responses. It is essentially a text conditioned image 2 image generation. That's why I tend to borderline reject this paper.

**Limitations:**

yes

**Quality:**

2

**Strengths And Weaknesses:**

Strengths:
1. the paper is well written and easy to follow.
2. the evaluation is complete with qualititative metrics (automatic and human based), quantitative images, ablation study, and memory efficiency analysis.

Weaknesses:
the main concern is that, the paper claims as a scene decorator, but the generation results are so different in terms of the background. It somehow totally change the background sturcture and does a re-creation from the input patches. My point is that, the generation consistency is not that good. and it's caused by the method design.

---

> ### Author Rebuttal · Authors · 2025-07-31
>
> We sincerely appreciate the time you dedicated to reviewing our paper and the insightful feedback you provided.
> In the following, we would like to present our point‑by‑point responses to your comments and concerns.
>
>
> ---
>
>
> > **Q1:** The main concern is that, the paper claims as a scene decorator, but the generation results are so different in terms of the background. It somehow totally change the background sturcture and does a re-creation from the input patches. My point is that, the generation consistency is not that good and it's caused by the method design.
>
> We appreciate the reviewer’s insightful concern and understand the concern regarding the changes in background observed in the generated results.
> Below, we would like to provide a clarification regarding our task setting.
>
> The task we study, scene-oriented story generation, aims to *maintain high-level scene consistency*, which involves preserving the environment, visual style, and overall atmosphere of a scene, rather than reproducing scenes at the pixel level. *Such a task definition aligns with the widely recognized objective of mainstream story generation*, allowing reasonable scene changes for contextual adjustments and adaptations to better fit the storylines, since scenes are not expected to remain completely static as the story changes.
>
> In other words, the essence of scene-oriented story generation lies in generating coherent and engaging story images, where variations in the scene are inevitably introduced to adapt to storytelling content. In this sense, our SceneDecorator is designed to function not as a low-level visual detail preserver but as a high-level visual semantic transmitter.
>
> If strict low-level visual preservation were enforced, the task would devolve into image editing or inpainting, which diverges from the objective of mainstream story generation.  In contrast, scene-oriented story generation preserves the core identity of the scene while allowing for controlled creative flexibility, which we believe is more aligned with real-world applications.
>
> Furthermore, we have conducted extensive experiments demonstrating that our approach achieves state-of-the-art performance, as validated across three aspects of evaluation: quantitative metrics, qualitative analysis, and user studies. These results showcase the effectiveness of our method in balancing scene coherence and adaptability in the context of story generation.
>
> We sincerely thank the reviewer for raising this critical point, and we hope this clarification addresses the concern.

---

> > ### Comment · Reviewer_ohEf · 2025-08-01
> >
> > Thanks for the author's response. I may have further quick questions:
> >
> > According to your response:
> > >Such a task definition aligns with the widely recognized objective of mainstream story generation, allowing reasonable scene changes for contextual adjustments and adaptations to better fit the storylines, since scenes are not expected to remain completely static as the story changes.
> >
> > One of the baselines you compared most is storydiffusion. From the storydiffusion we can see a figure of a **complete storyline**, which is composed of multiple images (>2), as you said in your rebuttal. It has a common people/object throughout the whole storyline. Although there are some changes, we can still figure out that it's telling a story.
> >
> > But throughout the main paper of SceneDecorator, I can't see any figures that demonstrate a complete story generated by this method. Can those image pairs be called by storylines? Maybe that is not so obvious at least for me. So could you explain a bit about the differences among (1) your work SceneDecorator, (2) text & image conditioned single image generation, (3) Storydiffusion (4) complete storylines generation?
> >
> > Thanks for your rebuttal again. I will increase my score if you can address my concerns.

---

> > > ### Author Response · Authors · 2025-08-02
> > > **Response to Reviewer ohEf**
> > >
> > > We sincerely appreciate your timely follow-up and insightful questions. In the following, we would like to present our point‑by‑point responses to your comments and concerns.
> > >
> > >
> > > ---
> > >
> > >
> > > > **Q1:** But throughout the main paper of SceneDecorator, I can't see any figures that demonstrate a complete story generated by this method. Can those image pairs be called by storylines? Maybe that is not so obvious, at least for me.
> > >
> > > We sincerely appreciate your insightful comment regarding the generated results in our paper.
> > >
> > > In addition to story images, corresponding story text is also essential to help readers better understand the storyline (or let's say, "feel" the storyline), just like the cases presented in Figure 1 of the StoryDiffusion paper. We apologize for not including the story text for all story images or only showing partial story text in our figures, as we aim to maintain simplicity and clarity for figures' visual elements and avoid excessive textual information.
> > >
> > > Below, we provide the story text for Scene-1 in Figure 1 as an example, to illustrate the storytelling capability of our method:
> > >
> > > 1. **Image 1**: "In a quiet corner of the abandoned courtyard, a wildflower blooms under soft light. A butterfly flutters in, lands gently, and sips nectar as the petals tremble."
> > >
> > > 2. **Image 2**: "A deer in the distance catches the motion. It lifts its head, ears high, and cautiously steps into the courtyard, scanning for hidden danger."
> > >
> > > 3. **Image 3**: "Soon after, a fox appears from behind the ruins. Catching the deer’s scent, it lowers its body and creeps forward, silent and focused."
> > >
> > > 4. **Image 4**: "The fox’s movement startles three squirrels on a fallen beam. One lets out a sharp squeak, and all three scatter upward."
> > >
> > > With the corresponding story text, the narrative of the story images for Scene-1 becomes clearer. Furthermore, it is also worth emphasizing that, as we claimed in our previous response, scene-oriented story generation focuses on maintaining high-level scene consistency. Compared to foreground (e.g., characters or objects) consistency, scene consistency is more difficult to perceive by humans and is often overlooked by previous studies. This challenge is precisely one of the motivations behind our research, as we believe the scenes, just like the sets arranged for filmmaking, also play a crucial role in visual storytelling.
> > >
> > > Once again, we are deeply grateful for your thoughtful feedback, and we will include these discussions along with the corresponding story text for the images in our final version of the paper to improve the completeness of our presentation.
> > >
> > >
> > > ---
> > >
> > >
> > > > **Q2:**  So could you explain a bit about the differences among (1) your work SceneDecorator, (2) text & image conditioned single image generation, (3) Storydiffusion (4) complete storylines generation?
> > >
> > > Thank you for allowing us to provide clarification. Using a more intuitive order, we present the definitions and highlight the differences below:
> > >
> > > 1. **Text & Image Conditioned Single Image Generation**: This task focuses on independently generating *one single* *image* aligned with the input textual (e.g., text prompt) or visual (e.g., depth map, sketch) conditions.
> > >
> > > 2. **Story Generation** (Complete Storylines Generation): This task aims to generate *multiple images* that maintain high-level concept consistency, with a series of text prompts (which together form a story) as input. Existing methods only consider foreground (e.g., characters or objects) consistency, meaning that the "concept" here typically refers to "foreground" in the initial scope of this task.
> > >
> > > 3. **StoryDiffusion**: It is a method of story generation, focusing on *foreground* *consistency*.
> > >
> > > 4. **SceneDecorator**: Our method is designed for scene-oriented story generation, which can be considered an extension of original story generation. This task focuses on *scene* *consistency*.
> > >
> > > We sincerely hope this clarification addresses your concern. To further improve the clarity of our paper, we would like to incorporate these discussions into our version.

---

> > > > ### Comment · Reviewer_ohEf · 2025-08-02
> > > >
> > > > Thanks for the responses from the authors, but I still have follow-up concerns:
> > > >
> > > > I respectfully disagree with your point that Figure 1 Scene 1 from Story 1 to Story 4 is a complete story. It seems that the authors tried to explain it as a complete story, but sadly, your naming pattern in your original paper reveals your original thought. In my understanding, **Every story-K is a separate story based on the input patch and text instead of a part of a story. It has no logical connections (temporal and causal) with prior images.** If the authors still strongly assert that those story-1, story-2, story-3, and story-4 are a single story, then please: (i) give me your "explaination of the story" for fig 1 scene 2 and 3. (ii) explain in your fig 2 caption in the middle: "Story-1: A fox explores the meadow, sniffing flowers under the moonlight.", "Story-2: A rabbit hops through the grass, seeking shelter for the night.", ..., and "Story-N: The girl dances among the trees, feeling the magic of the night." It is obvious that: **There are no logical connections between story-k and story-k+1**. Could the author give me any other comments that support your points in your response?
> > > >
> > > > If I'm not understanding wrongly, each story-k is a **text-image condition image generation**. SceneDecorator has a distinct difference from a complete storyline and storydiffusion. I tend to decrease my score if the author can not give me honest and reasonable responses.

---

> > > > > ### Author Response · Authors · 2025-08-03
> > > > > **Response to Reviewer ohEf**
> > > > >
> > > > > Thanks for your follow-up comment. In the following, we would like to present our point‑by‑point responses to your comments and concerns.
> > > > >
> > > > > ---
> > > > >
> > > > > > ***Q1**: I respectfully disagree with your point that Figure 1 Scene 1 from Story 1 to Story 4 is a complete story. It seems that the authors tried to explain it as a complete story, but sadly, your naming pattern in your original paper reveals your original thought.*
> > > > >
> > > > > According your previous comment, we may misunderstand your intention and think you are expecting a clearer illustration of the storyline in the generated story images. We apologize for this misunderstanding. Regarding the naming patterns, we would like to clarify that in the main paper, “Story-1” through “Story-N” refer to the 1st to Nth frames of the generated storyline.
> > > > >
> > > > > ---
> > > > >
> > > > > > ***Q2***: *There are no logical connections between story-k and story-k+1*
> > > > >
> > > > > StoryDiffusion generates a character-based storyline, where the high-level consistent character experiences different events in different scenes. We consider this a character-based logical connection, as the narrative follows the consistent character across different scenes.
> > > > >
> > > > > In contrast, SceneDecorator generates a scene-based storyline, where different characters experience different events within the high-level consistent scene. We consider this a scene-based logical connection, as the narrative follows the consistent scene across different characters, which is also acknowledged by **reviewer pX2S:** "a storyline involving different characters evolving within a consistent scene also represents a valid form of logical connection."
> > > > >
> > > > > Importantly, both methods yield similar storylines: StoryDiffusion follows the **consistent character across scenes**, while SceneDecorator follows the **consistent scene across characters**. Both storylines should be regarded as exhibiting logical connections.
> > > > >
> > > > > ---
> > > > >
> > > > > > ***Q3***: *Every story-K is a separate story based on the input* *patch* *and text instead of a part of a story. If I'm not understanding wrongly, each story-k is a text-image condition image generation. SceneDecorator has a distinct difference from a complete storyline and storydiffusion.*
> > > > >
> > > > > We respectfully **disagree** with your point that each story-*k* is a text-image conditional image generation.
> > > > >
> > > > > In practice, our proposed VLM-guided scene planning framework decomposes a global scene into several local scenes, and designs *K* story prompts for each local scene. Then, the generator takes a local-scene image and its corresponding *K* story prompts as input, generating *K* story images **simultaneously**. During the generation, we incorporate a **long-term scene-sharing** **attention** mechanism, which allows scenes across the *K* story images to **interact with each other**, ensuring high-level scene consistency.
> > > > >
> > > > > Importantly, our input-output format **matches** that of StoryDiffusion. Both methods use a reference image and *K* text prompts as input, simultaneously generating *K* story images. The main difference lies in the type of consistency each method maintains during the generation. By definition, both fall under the task of story generation.
> > > > >
> > > > > We sincerely hope the above responses can address your concerns.

---

### Official Review · Reviewer_pX2S · 2025-06-14

**Clarity:** 4
**Significance:** 4
**Originality:** 3
**Rating:** 6
**Confidence:** 5

**Summary:**

The paper introduces SceneDecorator, a novel training-free framework for scene-oriented story generation that addresses two key challenges: (i) scene planning to ensure narrative coherence across scenes, and (ii) scene consistency to maintain coherence across multiple storylines. Unlike existing methods that focus primarily on character consistency, SceneDecorator leverages a VLM-Guided Scene Planning strategy to decompose user-provided themes into globally coherent scenes and local sub-scenes in a "global-to-local" manner. Additionally, it employs a Long-Term Scene-Sharing Attention mechanism to preserve scene consistency while allowing subject style diversity. The framework demonstrates superior performance in qualitative and quantitative evaluations, with applications in creative domains like film and game design.

**Questions:**

None

**Ethical Concerns:**

["NO or VERY MINOR ethics concerns only"]

**Final Justification:**

I have read the comments of other reviewers.

After reviewing the responses from the authors, my concern is addressed and I think it is a good contribution for the research community.

Therefore, I have updated my final rating scores.

**Limitations:**

yes

**Quality:**

4

**Strengths And Weaknesses:**

### Strength

1. The technical approach is rigorous, combining VLMs (Qwen2-VL) and diffusion models (SDXL) innovatively. The proposed Scene-Sharing Attention and Extrapolable Noise Blending are well-designed to address long-term consistency with low computational overhead (Table 2).
2. The paper is well-structured, with clear motivations (e.g., limitations of independent scene generation in §1) and method descriptions (e.g., Figure 2 illustrates the pipeline effectively).
3. Addresses an underexplored problem (scene consistency in storytelling) with practical applications in film storyboarding and game design (§5).

### Weakness

1. Performance is bounded by the capabilities of FLUX.1-dev and Qwen2-VL
2. The evaluation relies on 32 users which may not capture diverse cultural interpretations of scene coherence.

---

> ### Author Rebuttal · Authors · 2025-07-31
>
> We sincerely appreciate the time you dedicated to reviewing our paper and the insightful feedback you provided.
> In the following, we would like to present our point‑by‑point responses to your comments and concerns.
>
>
> ---
>
>
> > **Q1:** Performance is bounded by the capabilities of FLUX.1-dev and Qwen2-VL.
>
> Thank you for raising this important point. While our method is  indeed built upon existing backbones, we would like to emphasize the **flexibility** of the designed paradigm.
>
> The core contribution of our work lies in the structured VLM-based scene planning framework and the long-term scene-sharing attention mechanism. These components are inherently plug-and-play and highly flexible, enabling seamless integration with both current and future foundation models without requiring architectural modifications. As demonstrated in Sec. 5 of the main paper, our approach is also compatible with a wide range of existing techniques, including PhotoMaker, ControlNet, and stylized LoRAs.
>
> Currently, we adopt FLUX.1-dev and Qwen2-VL as our base models, as they represent some of the state-of-the-art open-source models in their respective domains. We are confident that as advanced foundation models emerge, integrating them into our framework will further enhance performance and unlock greater potential.
>
> We hope this clarifies your concern, and we sincerely appreciate the opportunity to address this point in our final version.
>
>
> ---
>
>
> > **Q2:** The evaluation relies on 32 users which may not capture diverse cultural interpretations of scene coherence.
>
> We appreciate your valuable feedback and agree that broader and more diverse participants are important for evaluating cultural interpretations of generated results.
>
> To address this concern, we have expanded our user study to include an extended participant pool. Specifically, we conduct an extended evaluation involving participants from a range of countries, cultural backgrounds, genders, and age groups. This updated user study yields 61 valid responses, which we believe offer a more representative and comprehensive view of how Text Alignment, Scene Alignment, and Image Quality are perceived across different demographics. The results are summarized below:
>
> | **Methods**           | **Text Alignment** | **Scene Alignment** | **Image Quality** |
> | :-------------------- | :----------------: | :-----------------: | :---------------: |
> | CustomDiffusion       |        7.9%        |        3.4%         |       6.0%        |
> | Consistory            |       21.3%        |        14.1%        |       24.7%       |
> | StoryDiffusion        |       14.3%        |        6.3%         |       11.8%       |
> | SceneDecorator (Ours) |     **56.5%**      |      **76.2%**      |     **57.5%**     |
>
> These results show that our method, SceneDecorator, consistently outperforms existing baselines such as CustomDiffusion, ConsiStory, and StoryDiffusion, even when evaluated by participants from diverse cultural backgrounds. This superior performance further highlights the robustness and generalizability of our framework across cross-cultural users.
>
> We hope this expanded analysis helps address your concern, and the extended user study and its findings will be included in our final version.

---

> > ### Comment · Reviewer_pX2S · 2025-08-03
> > **New Question for Authors**
> >
> > The comments from other reviews have triggered a new thought for me.
> >
> > While I understand reviewer ohEf’s concern about the perceived lack of logical connection between story-k and story-k+1, this may stem from scene-centered narratives being less intuitive than character-centered ones. Nonetheless, I believe that, a storyline involving different characters evolving within a consistent scene, also represent a valid form of logical connection. This is often overlooked in prior work but carries practical value in commercial applications.
> >
> > But, I share the same confusion as reviewer ohEf: "Is each story-k a text-image conditioned image generation for independent generation?" If that is indeed the case, I believe it may be difficult to consider this as a truly consistent story generation.
> >
> > I hope the authors can address my confusion and provide a reasonable and objective justification.

---

> > > ### Author Response · Authors · 2025-08-03
> > > **Response to Reviewer pX2S**
> > >
> > > Thank you for the follow-up question. We sincerely appreciate your recognition of the logical coherence in our scene-based narrative storyline. In the following, we would like to present our point‑by‑point responses to your comments and concerns.
> > >
> > > ---
> > >
> > > > ***Q1***: *But, I share the same confusion as reviewer ohEf: "Is each story-k a text-image conditioned image generation for independent generation?" If that is indeed the case, I believe it may be difficult to consider this as a truly consistent story generation.*
> > >
> > > We would like to clarify that each story-k is **not** simply an independent text-image conditioned image generation.
> > >
> > > In practice, our proposed VLM-guided scene planning framework decomposes a global scene into several local scenes, and designs *K* story prompts for each local scene. Then, the generator takes a local-scene image and its corresponding *K* story prompts as input, generating *K* story images simultaneously. During the generation, we incorporate a **long-term scene-sharing** **attention** mechanism, which allows scenes across the *K* story images to **interact with each other**, ensuring high-level scene consistency.
> > >
> > > In addition, our input-output format aligns with that of StoryDiffusion. Both methods take a reference image and *K* text prompts as input, and simultaneously generate *K* story images. The key difference lies in the type of consistency each method preserves during generation. By definition, both fall under the task of story generation.
> > >
> > > We sincerely hope the above responses can address your concerns.

---

> > > > ### Comment · Reviewer_pX2S · 2025-08-04
> > > >
> > > > Thanks for the author's response. I have carefully evaluated the authors’ rebuttal and the comments from the other reviewers, and I think SceneDecorator is a meaningful and valuable work, and I have adjusted my score and confidence accordingly.
> > > >
> > > > I also read the comments from Reviewer ohEf, and I don't fully agree. In my view, a storyline that follows different characters within a consistent scene can still present a valid and coherent logical connection. Such narrative design, which maintains scene consistency, is both meaningful and technically justified. I think this concern might stem from limited familiarity with this subfield.
> > > >
> > > > After reviewing the authors’ latest rebuttal and revisiting the paper, my concern is resolved, as the consistency is achieved through Long-Term Scene-Sharing Attention rather than simple single-image generation. And the experimental results further demonstrate that this work effectively addresses a critical gap in story generation models (e.g., compared to StoryDiffusion, Consistory). This is a well-executed contribution that aligns with the standards of NeurIPS, and I believe it deserves acceptance.

---

> > > > > ### Author Response · Authors · 2025-08-04
> > > > > **Response to Reviewer pX2S**
> > > > >
> > > > > Once again, we sincerely thank you for taking the time to review our rebuttal and for acknowledging that "a storyline that follows different characters within a consistent scene can still present a valid and coherent logical connection." We truly appreciate your recognition of our contribution and experimental results, and are very pleased that our responses have addressed your confusion regarding the scene-consistent story generation.

---

> ### Comment · Reviewer_pX2S · 2025-08-02
>
> Thank you for your detailed response. Your rebuttal address my concerns. I think this is a good paper and consider to update my confidence about my judgment.

---

### Official Review · Reviewer_pRBq · 2025-07-01

**Clarity:** 3
**Significance:** 2
**Originality:** 3
**Rating:** 4
**Confidence:** 3

**Summary:**

The paper introduces a training-free framework, scenedecorator, that aims to impose coherence in scenes, with subject style diversity across different stories; essentially a scene-oriented story generation instead of character-oriented story generation widely available in the literature.

**Questions:**

What do the authors think about extending from scene-oriented story generation to scene-oriented movie generation? What would be the challenges and technical problems to solve?

**Ethical Concerns:**

["NO or VERY MINOR ethics concerns only"]

**Final Justification:**

I'm happy with accepting the paper, although I won't champion for it.

**Limitations:**

Not exactly a limitation of this work, but my question would be about the next steps. From story to animation movie generation!

**Paper Formatting Concerns:**

Good length, well-organized and nicely presented paper. No complains about the formatting.

**Quality:**

3

**Strengths And Weaknesses:**

Scene centric story generation and scene consistencies across subject diversity - these two aspects are the focus of the work. The authors propose to perform mask-guided scene injection and scene-sharing attentions to achieve the above mentioned goals. The qualitative examples help grasp the idea behind the work easily.
This is an interesting and possibly useful tool for content creators.

---

> ### Author Rebuttal · Authors · 2025-07-31
>
> We sincerely appreciate the time you dedicated to reviewing our paper and the insightful feedback you provided.
> In the following, we would like to present our point‑by‑point responses to your comments and concerns.
>
>
> ---
>
>
> > **Q1:** What do the authors think about extending from scene-oriented story generation to scene-oriented movie generation? What would be the challenges and technical problems to solve?
>
> We appreciate the reviewer’s forward-looking insight. Extending from scene-oriented story generation to scene-oriented movie generation is a valuable and exciting direction with significant potential to push the boundaries of AI-driven visual content creation. In the following, we discuss potential challenges and technical problems, respectively.
>
> ### **Challenges**
>
> Extending from scene-oriented story generation to movie generation is a natural and promising direction, but it also introduces new technical demands. Current methods rely on image generation models to handle isolated scenes. Moving toward movie generation requires video generation models that can account for temporal dynamics and maintain consistency over time. Based on our observations, we outline three challenges worth exploring:
>
> 1. **Multi-Scene Shot Transitions:** In a movie, transitions between shots and scenes also carry narrative expression. How a scene begins and ends, how camera angles shift, and how characters move across shots all affect storytelling. One key challenge is making these transitions smooth and logically consistent so that the final result doesn’t feel fragmented. This will likely require finer control over visual continuity and temporal alignment.
> 2. **Scene-Expandable Movie Generation**: Generating movies where scenes can dynamically expand or evolve depending on the narrative context presents another challenge. For instance, enabling the system to transition from focused, character-driven moments to broader, environment-focused shots requires the ability to generate multi-scale content while maintaining narrative cohesion. This capability can support richer storytelling and more immersive visual experiences.
> 3. **Multi-Modal Integration with Scenes**: Combining scene information with multi-modal conditions, such as synchronized audio (e.g., dialogue, ambient sounds) or textual descriptions, is key to enriching storytelling. This requires aligning visual semantics with auditory and textual elements while ensuring temporal consistency. Effective integration across modalities can greatly enhance the emotional depth and immersion of generated movies.
>
> ### **Technical Problems**
>
> Naturally, scene-oriented movie generation still faces several technical limitations due to the increased complexity of generating temporally consistent, high-quality video content compared to static images. Below, we highlight two critical technical problems that require further exploration:
>
> 1. **Computational Cost:**  Movie generation is inherently more resource-intensive than image generation, as it involves producing high-resolution video frames with consistent temporal dynamics. This leads to significantly higher computation, memory, and storage requirements. To address these issues, methods such as model compression and sparse computation could be explored to reduce computational overhead while ensuring scalability during training and inference.
> 2. **Generation Quality:** Ensuring generation quality is crucial, as it directly impacts the audience's engagement and the overall storytelling experience. High-quality movie generation requires maintaining narrative coherence, stylistic consistency, and a balance between creativity and realism. The system must ensure that visual elements align with the storyline, follow a cohesive artistic direction, and produce compelling yet believable content.
>
> We believe that scene-oriented movie generation will emerge as an important research area in the near future, and we hope our work can serve as a foundation for further exploration in this domain. Once again, we sincerely thank the reviewer for the insightful comments, which have inspired us to think deeply about the future directions of our research.

---

> > ### Comment · Reviewer_pRBq · 2025-08-07
> > **story to movie insights**
> >
> > Given the unprecedented success of controllable video generation, e.g. image conditioned generation, camera and motion control, even world models - I was expecting a deeper insights with practical challenges.
> >
> > However, I understand that movie generation isn't the focus of the work and it can't penalized for lack of understanding for something beyond the scope.

---

> > > ### Author Response · Authors · 2025-08-08
> > > **Response to Reviewer pRBq**
> > >
> > > We sincerely appreciate your thoughtful feedback. Indeed, movie generation lies beyond the scope of our current work. Nonetheless, we greatly value your insightful comments and recognize the importance of the mentioned techniques. As you suggested, we will certainly consider exploring these directions in our future research endeavors.

---

### Official Review · Reviewer_YYxn · 2025-07-03

**Clarity:** 2
**Significance:** 3
**Originality:** 2
**Rating:** 5
**Confidence:** 4

**Summary:**

SceneDecorator is a training-free framework designed for scene-oriented story generation, aiming to maintain visual consistency across a series of images. It addresses the limitations of existing methods that the paper claims often focus on character consistency while neglecting the role of scenes in storytelling

**Questions:**

1. Could authors please provide a precise description of the modifications made to each baseline (CustomDiffusion, ConsiStory, and StoryDiffusion)?
2. How robust is the VLM planning stage? What is the approximate failure rate (e.g., instances where the VLM provided unusable coordinates, nonsensical storylines, or failed to adhere to the theme)?
3. Can "SceneDecorator" be called as   Scene-Oriented Story Generation in light of its inability to handle stories with scenes that evolve or change over time (e.g., a landscape changing from day to night, a location being altered by character actions)?

**Ethical Concerns:**

["NO or VERY MINOR ethics concerns only"]

**Final Justification:**

The authors have successfully addressed key concerns regarding experimental rigorand responsible research practices but their response to the system's core conceptual limitation (Q4) is no promising.

Overall, I think this paper should be accepted.

**Limitations:**

The authors have not adequately addressed the potential negative societal impact of their work. In the NeurIPS Paper Checklist, when asked if the paper discusses potential positive and negative societal impacts, the authors answered "[NA]" (Not Applicable). However, this dismissal is inadequate. A tool that improves the generation of consistent, multi-image narratives could be used to create more convincing and coherent disinformation or propaganda.

**Quality:**

3

**Strengths And Weaknesses:**

Strengths:
1. The Problem story consistency is a genuine problem, and the authors focus on scenes instead of just characters is a valid angle.
2. The qualitative results in the figures are undeniably attractive.
3. The core idea of using a powerful VLM to do the "thinking"—planning the global scene, then dicing it up into local scenes and storylines—is a pragmatic engineering choice

Weaknesses：
1. The “VLM-Guided Scene Planning” module is described as the core novelty of the pipeline. However, it appears to be primarily a well-crafted prompting mechanism applied to a powerful pretrained VLM (Qwen2-VL), with little algorithmic innovation. The formulation (e.g., Q = Fₜ(T)) may overstate the underlying complexity.
2. The entire "global-to-local" pipeline is built on a fatal assumption: that a story's scenes are just different crops of a single, static master image. This is a profound limitation. This framework is incapable of depicting scenes that evolve or change over time, which is a core element of countless narratives.

---

> ### Author Rebuttal · Authors · 2025-07-31
>
> We sincerely appreciate the time you dedicated to reviewing our paper and the insightful feedback you provided.
> In the following, we would like to present our point‑by‑point responses to your comments and concerns.
>
>
> ---
>
>
> > **Q1:** The “VLM-Guided Scene Planning” module is described as the core novelty of the pipeline. However, it appears to be primarily a well-crafted prompting mechanism applied to a powerful pretrained VLM (Qwen2-VL), with little algorithmic innovation. The formulation (e.g., Q = Fₜ(T)) may overstate the underlying complexity.
>
> We appreciate the reviewer’s attention to the innovation of VLM-Guided Scene Planning. Our method indeed builds upon the pretrained Qwen2-VL model, but we would like to emphasize that our approach equips VLMs with the capability of scene planning, which they do not inherently possess.
>
> Specifically, our innovation does not lie in modifying the model architecture itself, but in *the deliberate design of a set of strategies for achieving scene planning via VLMs*, which includes:
>
> 1. **conceptualizing the global scene**, where the VLM generates a comprehensive and cohesive scene description based on a user-provided theme;
> 2. **visualizing the global scene**, using an excellent T2I generator to transform these descriptions into high-quality visual representations; and
> 3. **crafting local scenes and stories**, where the global scene is decomposed into multiple local sub-scenes and narratives to ensure both semantic coherence and narrative continuity.
>
> These strategies collectively empower VLMs to undertake scene planning, transforming them from general-purpose models into specialized ones capable of bridging the gap between abstract themes and semantically coherent visual storytelling, which is a capability not previously explored in existing methods.
>
> As mentioned above, our method adopts a three-stage process for scene planning. To precisely explain the functionality of each stage, we represent each stage as a series of formalized functions in our paper. These formulations are not intended to overstate the complexity but to explicitly clarify the distinct inputs and outputs at each stage, thereby providing a more systematic and informative illustration of the overall workflow.
>
> In summary, our method introduces a novel way of utilizing VLMs for scene planning by designing a three-stage process that includes conceptualizing the global scene, visualizing the global scene, and crafting local scenes and stories. This combination of strategies represents a significant step forward in achieving scene-oriented story generation, exploring a technical solution that has not been studied by existing methods.
>
>
> ---
>
>
> > **Q2:** Could authors please provide a precise description of the modifications made to each baseline (CustomDiffusion, ConsiStory, and StoryDiffusion)?
>
> Thank you for the question regarding the implementation details of the baselines. Below, we clarify how each method is adapted to our task setting:
>
> 1. **CustomDiffusion** is a tuning-based method that requires fine-tuning to inject given visual concepts. Specifically, we fine-tune the CustomDiffusion using the local scene images obtained from our VLM planning module as training data. During inference, we adopt the corresponding story sub-prompts together with the fine-tuned model to enable a comparison under our task setting.
> 2. **StoryDiffusion** is a training-free method designed for subject-consistent story generation and naturally supports image conditioning. We directly use the local scene images and the associated story sub-prompts from the VLM planning module as input for comparison.
> 3. **ConsiStory** is also a training-free method for subject-consistent story generation. However, the official implementation does not support images as input. For a meaningful comparison, we make minimal modifications to adapt it to our task, prepending the local scene images to the input batch and leveraging it as a reference through self-attention and feature infusion mechanisms. Additionally, the same story sub-prompts are also adopted accordingly.
>
> It is also worth noting that all experiments are conducted on the same scene dataset processed by the VLM planning module to ensure a fair comparison.
>
> We hope this resolves your concern, and we sincerely appreciate the opportunity to include these descriptions in our final version.
>
>
> ---
>
>
> > **Q3:** How robust is the VLM planning stage? What is the approximate failure rate (e.g., instances where the VLM provided unusable coordinates, nonsensical storylines, or failed to adhere to the theme)?
>
> We sincerely thank the reviewer for this insightful comment. Accordingly, we would like to discuss the robustness of VLM-Guided Scene Planning in two aspects:
>
> 1. **Coordinate Rationality:** Failures where the VLM generates unusable coordinates are exceptionally rare. To seamlessly handle such cases, we have implemented a simple yet effective correction strategy: invalid coordinates are automatically adjusted by snapping them to the nearest valid bounding box, which ensures proper cropping.
> 2. **Semantic Rationality:** To evaluate the success rate of producing semantically reasonable scenes, we conducted a quantitative assessment using GPT-4o. Specifically, for each sample, we input the story theme, the global scene, the obtained local scenes, and the story sub-prompts into GPT-4o, which is guided to conduct evaluation across three key criteria: Narrative Coherence, Theme Adherence, and Layout Reasonableness. Each criterion was scored on a 10-level scale, with scores ranging from 0 to 100. The evaluation was performed on 2,920 samples, showing that our method can also achieve satisfactory performance in the semantic aspect, with the results summarized below:
>
> | **Narrative Coherence** | **Theme Adherence** | **Layout Reasonableness** |
> | :---------------------: | :-----------------: | :-----------------------: |
> |         90.06%          |       92.57%        |          90.29%           |
>
> In summary, our VLM-Guided Scene Planning shows strong robustness in both coordinate and semantic rationality, highlighting its effectiveness in producing coherent and reasonable scenes for visual storytelling.
>
> We will incorporate these discussions and quantitative results into our final version to provide a more comprehensive analysis.
>
>
> ---
>
>
> > **Q4:** The entire "global-to-local" pipeline is built on a fatal assumption: that a story's scenes are just different crops of a single, static master image. This is a profound limitation. This framework is incapable of depicting scenes that evolve or change over time, which is a core element of countless narratives. / Can "SceneDecorator" be called as Scene-Oriented Story Generation in light of its inability to handle stories with scenes that evolve or change over time (e.g., a landscape changing from day to night, a location being altered by character actions)?
>
> We thank the reviewer for raising this important question regarding the temporal dynamics of scenes in our framework. In fact, our SceneDecorator is capable of handling stories with scenes that evolve or change over time, through two distinct capabilities integrated within our design:
>
> 1. **Creating time-evolving story sub-prompts:** Given a local scene image, our VLM-guided scene planning module can generate a series of story sub-prompts that unfold over time within that scene (e.g., a local detail being altered by character actions), rather than producing a static story description. In this way, the VLM can infer a sequence of events based on its commonsense knowledge, such as a character arriving, interacting with the environment, and leaving, all within the same scene context. These story sub-prompts can then be used by the subsequent image generator to produce story images that change over time, achieving visual storytelling with temporal evolution.
> 2. **Creating time-evolving story scenes:** Alternatively, we can also generate multiple global scene images that evolve over time (e.g., a landscape shifting from day to night) in the VLM-guided scene planing module. The VLM can then collaboratively plan and crop local scenes simultaneously from multiple global scenes to create a time-evolving storyline. The resultant local scenes naturally inherit the time-evolving visual semantics from their respective global scenes, enabling the generation of visual stories that reflect changes over time.
>
> This flexibility allows SceneDecorator to support dynamic, time-evolving narratives, thereby addressing the mentioned limitations. Due to the policy prohibiting the presentation of images during the rebuttal phase, we will present the relevant experiments and visual results, along with these discussions, in our final version.
>
>
> ---
>
>
> > **Q5:** The authors have not adequately addressed the potential negative societal impact of their work.
>
> We sincerely appreciate your valuable feedback and apologize for initially overlooking the potential societal impacts of our work.
>
> Our work is designed for scene-oriented story generation in the domains of visual content creation. However, we recognize that any technology capable of generating multi-image narratives may carry societal risks if misused. Specifically:
>
> 1. **Disinformation and propaganda**: The ability to generate visual narratives could be exploited to create persuasive but false stories, amplifying the spread of disinformation.
> 2. **Bias and stereotypes**: Unintended biases in input story theme could lead to the reinforcement of harmful stereotypes or discriminatory representations.
> 3. **Inappropriate or harmful content**: Without proper safeguards, generated content might unintentionally include sensitive, violent, or inappropriate material, potentially causing emotional harm to audiences.
>
> As you suggested, we will carefully re-check and revise our checklist response and include a thorough discussion of societal impacts in our final version.

---

> > ### Comment · Reviewer_YYxn · 2025-08-08
> >
> > Thank you very much for your rebuttal.
> >
> > Only for Q4, The mechanisms described in the rebuttal, particularly the simultaneous planning and cropping from multiple global scenes, seem far more complex than the "global-to-local" pipeline described in the original paper.  This feels less like a clarification and more like a brand-new capability introduced during the rebuttal to patch a fundamental flaw. Without any evidence in the current paper or rebuttal, this extraordinary claim is entirely unsupported.
> >
> > The authors have successfully addressed other concerns. My overall score would increase.

---

> > > ### Author Response · Authors · 2025-08-08
> > > **Response to Reviewer YYxn**
> > >
> > > Thank you very much for your insightful feedback and for increasing your score. Regarding Q4, the mechanisms we described are indeed technically feasible within our existing framework, requiring only minor adjustments to the VLM's prompt. Due to the rebuttal policy prohibiting the presentation of images, we commit to providing the corresponding experimental results along with the setup in the final version to support our claims. We greatly appreciate your understanding and your positive evaluation of our efforts.

---

### Decision · Program_Chairs · 2025-09-17

**Decision:**

Accept (poster)

**Comment:**

Proposed approach is designed to generate multi-image stories with consistent background and context. The reviews for the paper are largely positive with 1 x Borderline Accept, 1 x Accept, 1 x Strong Accept and 1 x Borderline Reject. Reviewers raised a number of cenrns regarding the work, including (1) with assumptions that story scenes are different crops of the single static master image [YYxn], (2) bounds on performance that are underlined by FLUX.1-dev and Qwen2-V [pX2S] and (3) lack of generation consistency in the results [ohEf].Authors have provided responses to these concerns. In general reviewers were convinced with the responses with exception of [ohEf], who remains concerned about consistency of characters among the panels of the story.

AC has read the reviews, rebuttal and the discussion that followed. AC agrees with [ohEf] that there is a lack of consistency with respect to characters. However, AC also agrees with authors that the focus of the paper is on the scene consistency, where indeed the proposed approach excels. Ideally, it would be nice to see both, which AC believes would be possible if scene guidance (proposed) was to be combined with character guidance (prior work) for a fully consistent result. Unfortunately, the paper falls short of this. At the same time, the proposed approach is still valuable and likely to motivate future work where scene and character consistency are both preserved. On these grounds AC is recommending Acceptance.